# Downregulation of Histone H3.3 Induces p53-Dependent Cellular Senescence in Human Diploid Fibroblasts

**DOI:** 10.3390/genes15050543

**Published:** 2024-04-25

**Authors:** Yuki Yamamoto, Ryou-u Takahashi, Masaki Kinehara, Kimiyoshi Yano, Tatsuya Kuramoto, Akira Shimamoto, Hidetoshi Tahara

**Affiliations:** 1Department of Cellular and Molecular Biology, Basic Life Sciences, Institute of Biomedical and Health Sciences, Hiroshima University, Hiroshima 734-8553, Japan; yyamamoto@hiroshima-u.ac.jp (Y.Y.); rytakaha@hiroshima-u.ac.jp (R.-u.T.);; 2Laboratory of Genome Stress Signaling, National Cancer Center Research Institute, Tokyo 104-0045, Japan; kiyano@ncc.go.jp; 3Faculty of Pharmaceutical Sciences, Sanyo-Onoda City University, Sanyo Onoda 756-0884, Japan; shim@rs.socu.ac.jp

**Keywords:** histone, p53, aging, senescence, microRNA

## Abstract

Cellular senescence is an irreversible growth arrest that acts as a barrier to cancer initiation and progression. Histone alteration is one of the major events during replicative senescence. However, little is known about the function of H3.3 in cellular senescence. Here we found that the downregulation of H3.3 induced growth suppression with senescence-like phenotypes such as senescence-associated heterochromatin foci (SAHF) and β-galactosidase (SA-β-gal) activity. Furthermore, H3.3 depletion induced senescence-like phenotypes with the p53/p21-depedent pathway. In addition, we identified miR-22-3p, tumor suppressive miRNA, as an upstream regulator of the *H3F3B* (H3 histone, family 3B) gene which is the histone variant H3.3 and replaces conventional H3 in active genes. Therefore, our results reveal for the first time the molecular mechanisms for cellular senescence which are regulated by H3.3 abundance. Taken together, our studies suggest that H3.3 exerts functional roles in regulating cellular senescence and is a promising target for cancer therapy.

## 1. Introduction

Cellular senescence is a physiological and homeostatic processes with permanent cell growth arrest and is induced by a wide range of cellular stresses such as DNA damage, oxidative stress, and oncogene activation [1,2,3]. Senescent cells are characterized by flattened morphology, heterochromatic changes, loss of Lamin-B1 and, the expression of senescence-associated ß-galactosidase (SA-βGal). In most cases, tumor suppressor genes such as p53/p21 and p16 are activated in senescent cells. Therefore, cellular senescence is considered to act as a natural barrier to tumorigenesis [1,2,3,4,5,6].

The senescence program is activated and maintained by multiple pathways [7] such as p53, retinoblastoma protein (RB), and p16^INK4a^. Immortal cells arise due to defects or mutations of genes involved in these pathways [8]. Some studies have indicated that the senescence phenotype is accompanied by senescence-associated heterochromatin foci (SAHF), which leads to extensive stable chromatin changes and an irreversible growth arrest [9,10,11]. The chromatin changes are accompanied by a reduction in the expression levels of multiple histones, including CENP-A, H2B, H3, H4, and linker histone H1 during senescence [12,13,14,15]. On the other hand, histone variants, H3.3 and macroH2A, are deposited into chromatin of replicative or oncogene Ras-induced senescent cells [15,16,17].

Histone variant H3.3, which is encoded by two genes, *H3F3A* and *H3F3B*, is enriched at nucleosomes at transcription start sites, enhancers, and intragenic regions of actively transcribed genes [18,19,20]. On the other hand, H3.3 is also enriched at silent loci in pericentric heterochromatin and telomeres [20]. Recently, it has been reported that H3.3 levels are increased during senescence, and that the overexpression of H3.3 or an H3.3 cleavage product, which lacks the first 21 amino acids in its N-terminal region, induces senescence in IMR90 fibroblasts [21]. In addition, Pak2 induced HIRA-mediated nucleosome assemblies of H3.3 in human fibroblasts during oncogenic-induced senescence [22]. On the other hands, loss of CDH1 reduced H3.3 levels in brain chromatin, which leads to lifespan shortening [23]. While a number of studies report the association of H3.3 expression with cellular senescence and aging, the roles of H3.3 in cellular senescence still remain unclear.

Recently, some studies have demonstrated the importance of microRNAs for cellular senescence during tumor suppression [24,25,26,27]. MicroRNAs are classified as small non-coding RNAs that negatively regulate gene expression through cleavage or translation repression of target transcripts [28,29,30]. Generally, microRNAs bind to the 3′-untranslated region (3′-UTR) of their mRNA targets by using the seed region at the 5′ end of the microRNA [29,30,31,32], and they appear to directly regulate a number of mRNA targets, which are involved in various biological processes, such as cell differentiation, proliferation, tumorigenesis, apoptosis, and senescence. We previously reported that miR-22 is upregulated in human senescent fibroblasts, and that miR-22 overexpression induces cellular senescence through the downregulation of *CDK6*, *SIRT1*, and *SP1* genes, all involved in the senescence program [25]. In addition, synthetic miR-22 mimic delivery suppresses tumor growth and metastasis through the induction of cellular senescence in a mouse model of breast cancer [25]. Although miR-22 has two mature products, miR-22-5p and miR-22-3p, the selection of functional miR-22 strands and its replicative senescence-associated target genes are not fully defined.

In this study, we focused on the roles of H3.3 in replicative senescence using human fibroblasts. We report here that the downregulation of H3.3 induced p53-dependent cellular senescence in human fibroblasts. Depletion of H3.3 by specific siRNAs against *H3F3A* and *H3F3B* genes resulted in growth arrest, SAHF formation, and increased senescence-associated β-galactosidase (SA-β-gal) activity, accompanied by activation of p53. Furthermore, we also found that miR-22-3p, but not miR-22-5p, directly targeted the 3′-UTR of *H3F3B* mRNA, encoding for H3.3 protein. Our findings, therefore, provide novel insights into the critical roles of H3.3 for senescence induction and the molecular mechanisms by which H3.3 expression is downregulated during cellular senescence.

## 2. Materials and Methods

### 2.1. Cell Culture

TIG-3 and 293T cells were cultured in DMEM (Sigma-Aldrich, St. Louis, MO, USA) supplemented with 10% FBS (vol/vol) (Thermo Fisher Scientific, Waltham, MA, USA), and 0.5× antibiotic-antimycotic liquid (Gibco, Waltham, MA, USA).

### 2.2. RT-qPCR

Total RNAs were purified from young (44–60 PDL), pre-senescent (62–71 PDL), and senescent (72–75 PDL) TIG-3 cells by using miRNeasy kit (QIAGEN, Hilden, Germany) and quantified with a spectrophotometer (NanoPhotometer; IMPLEN, München, Germany). cDNAs were synthesized using high-capacity RNA-to-cDNA Kit (AppliedBiosystems, Waltham, MA, USA) or miRCURY LNA universal RT microRNA system (Exiqon, Vedbaek, Denmark). Real-time qRT-PCR was performed with a real-time PCR system (Rotor-Gene; QIAGEN) by using miRCURY locked nucleic acid (LNA) primers for miR-22-3p (Exiqon) and U6 (Exiqon). The expression of *H3F3A*, *H3F3B*, and *GAPDH* was detected with primers (for *H3F3A*, 5′-AAAAATAGGGGACAGAAATCAGG-3′ and 5′-TAGTGAATGGATGGAATCTACTG-3′, for *H3F3B*, 5′-AATAGTGCTGTATTTGCAGTGTGG-3′ and 5′-AGATACCTACTTTACCTTCCCTCC-3′, for *GAPDH*, 5′-CCTCTGACTTCAACAGCGACAC-3′ and 5′-GCCAAATTCGTTGTCATACCAG-3′). Relative expression levels were calculated using the 2^−ΔΔCt^ method after normalization with reference to the expression levels of U6 small nuclear RNA or GAPDH.

### 2.3. DNA Microarray

The data utilized for comparison of histone RNA expression between young and senescent TIG-3 were acquired from Gene Expression Omnibus (GEO) database (Accession number GSE162201).

### 2.4. Western Blotting

Cells were lysed with 2× sample buffer (120 mM Tris-HCl, pH6.8, 4% SDS, 200 mM DTT, 0.004% Bromophenol Blue, and 12% glycerol). The lysed proteins were separated by SDS-PAGE and transferred to polyvinylidene fluoride membrane. Antibodies to p53 (sc-126, Santa Cruz biotechnology, Dallas, TX, USA), phosphorylated p53 (ser15; 9284, Cell Signaling Technology, Inc., Danvers, MA, USA) phospho-Rb (ser807/811; 9308, Cell Signaling Technology), β-actin (A5441, Sigma-Aldrich), and histone H3.3 (09-838, Merck Millipore, Billerica, MA, USA) were purchased. The secondary antibodies were HRP-conjugated anti-rabbit (111-035-003, Jackson ImmuneResearch Laboratories, Inc., West Grove, PA, USA) and anti-mouse (115-035-003, Jackson ImmuneResearch Laboratories, Inc.) antibodies. Immunoreactive bands were visualized by using western lightning plus ECL (GE Healthcare, Chicago, IL, USA), followed by exposure to imageQuant LAS 4000, a digital imaging system (GE Healthcare). The chemiluminescence signal intensity was calculated using ImageJ software (version 1.52q).

### 2.5. Luciferase Reporter Assay

The full-length 3′-UTRs (WT) of human *H3F3B* containing three putative miR-22-3p binding sites were amplified by PCR with primer pair (hH3F3B_470U, 5′-TATGCTAGCATCCACGCTAAGAGAGTCACCATC-3′ and hH3F3B_2652L, 5′-TATGTCGACCCATCAAGGGCATAGGATACCTGC-3′) from genomic DNA and cloned into the NheI and SalI sites on pmirGLO vector (Promega, Madison, WI, USA). pmirGLO vector has one insertion site of miRNA target sequence at the 3′ UTR of the firefly luciferase gene. Renilla luciferase gene was also coded into pmirGLO vector for normalization. A series of vectors with deletions of the putative miR-22-3p binding site 1 and 2 were constructed by inverse PCR with primer pairs (hH3F3B_Δ1029-1035U, 5′-ATAGAATACACTATGTGCATTTATAATAGC-3′, hH3F3B_Δ1029-1035F, 5′-AGGTTTACTTTTTTTTTTTTTAAAGG-3′, hH3F3B_Δ1369-1374U, 5′-TTTCATTGTGTTGTGTGGTTGG-3′, and hH3F3B_Δ1369-1374F, 5′-ACTAACTAGTTCAGAATGTTAGTTAAGATG-3′) and KOD-plus-mutagenesis kit (TOYOBO, Osaka, Japan). The vectors deleted the site 1–2 and 3 regions were constructed by SacI and AccI digestion of the full-length WT 3′-UTR vector, respectively. All constructs were confirmed by DNA sequencing.

For 3′-UTR luciferase reporter assays, each construct (100 ng) was co-transfected with miR-22-3p mimic or miR-control mimic (mirVana miRNA, Thermo Fisher Scientific) into a 96-well plate with DharmFECT Duo transfection reagent (Thermo Fisher Scientific). Luciferase assays were performed with the dual-luciferase reporter system (Promega) according to the manufacturer’s instruction at 72 h after transfection. Luminescent signal was quantified by luminometer (Glomax; Promega), and each value from firefly luciferase constructs was normalized by renilla luciferase assay.

For p21 promoter-driven luciferase reporter assays, the p21 promoter region of human *CDKN1A* containing p53 binding sites [33,34] was amplified by PCR with primer pair (p21-p53RE-Sfil-F, 5′-GGCCTAACTGGCCCACCACTGAGCCTTCCTCAC-3′ and p21-p53RE-Sfil-R, 5′-GGCCGCCGAGGCCCTGACTCCCAGCACACACTC-3′) from genomic DNA and cloned at the SfiI sites into pGL4.23 vector (Promega). The construct was transfected with miR-22-3p mimic or miR-control mimic using a square electric pulse generating electroporator, NEPA21 (Nepa Gene, Chiba, Japan), and luciferase assays were performed with the dual-luciferase reporter system (Promega) at 72 h after transfection according to the manufacturer’s instructions.

### 2.6. miRNA/siRNA Transfection

Cells were reverse-transfected with 10 nM of miRNA and/or siRNA with Lipofectamine RNAiMax (Invitrogen, Carlsbad, CA, USA) according to the manufacturer’s protocol. miRNAs (all mirVana™ miRNA), negative-control miRNA (Cat.# 4464058) and miR-22-3p (Cat.# 4464066, MC16232) mimic were obtained from Thermo Fisher Scientific. siRNAs targeting *TP53* (Thermo Fisher Scientific, Stealth RNAi™ siRNA Cat.# 4390824, s605), *H3F3A* (GE Healthcare, siGENOME SMARTpool, Cat.#M-011684-01-0005), *H3F3B* (GE Healthcare, siGENOME SMARTpool, Cat.#M-012051-00-0005), and negative-control siRNA (Cat.#4390843 or D-001206-13-05) were purchased from Thermo Fisher Scientific and GE Healthcare.

### 2.7. SA-β-Gal Assay

For detection of SA-β-gal activity, SA-β-gal staining was performed as described previously [35,36]. Five days after transfection, SA-β-gal-positive cells were stained and quantified by counting positive and negative cells in at least five randomly chosen independent fields. Pictures were taken with a 10× phase-contrast objective on a light microscope (IMT-2; Olympus, Tokyo, Japan).

## 3. Results

### 3.1. Histone Variant H3.3 Was Decreased during Replicative Senescence in Human Diploid Fibroblasts

Microarray analysis using young and senescent human fibroblast cells, TIG-3 cells revealed that the expression levels of both *H3F3A* and *H3F3B* genes were drastically decreased during replicative senescence in TIG-3 cells (Figure 1A,B). Consistent with the results of microarray analysis, RT-qPCR showed the downregulation of *H3F3A* and *H3F3B* genes in senescent TIG-3 cells compared to young TIG-3 cells (Figure 1C). Furthermore, the protein abundance of H3.3 was similarly decreased during replicative senescence (Figure 1D).

It has been known that p53 directly regulates the expression levels of p21, a cyclin-dependent kinases (CDKs) inhibitor, and promotes cellular senescence [7,33]. P21 inhibits the phosphorylation of RB through CDKs inhibition, followed by cell growth arrest [7,33]. Therefore, in addition to H3.3 expression, we also examined the p53, p21 expression and Rb phosphorylation using young and senescent TIG-3 cells and confirmed the inhibition of RB phosphorylation at ser-807/811 and upregulation of p21 expression only in senescent TIG-3 cells (Figure 1D).

### 3.2. H3.3 Depletion Induced Cellular Senescence through p53/p21 Activation

To investigate H3.3 functions in cellular senescence, we performed the depletion of H3.3 in young TIG-3 cells using specific siRNAs against *H3F3A* and *H3F3B*. H3.3 knockdown was confirmed by western blotting analysis (Figure 2A). Interestingly, the knockdown of H3.3 induced the expression levels of p21 and inhibited the phosphorylation of RB at ser-807/811 (Figure 2A). However, the H3.3 depletion did not alter the expression levels of p53 or ser-15 phosphorylation (Figure 2A).

To further examine whether the transcriptional activity of p53 is enhanced by H3.3 depletion, a p21 promoter-driven luciferase reporter, containing p53 binding sites, was tested. p21 promoter activity was upregulated by depletion of H3.3 as well as by doxorubicin, a p53 activator, compared with DMSO or siRNA controls (Figure 2B). In young TIG-3 cells, H3.3 depletion inhibited cell growth (Figure 2C) and increased the formation of SAHF (Figure 2D). Thus, these results suggest that H3.3 depletion induced p53 stabilization and/or activation, which resulted in cellular senescence.

We also analyzed the effects of silencing p53 and/or H3.3 by specific siRNAs on SA-β-gal activity, a well-known senescence marker [35]. Silencing of *p53, H3F3A,* and *H3F3B* by siRNAs increased the SA-β-gal-positive in young TIG-3 cells (Figure 2F,G). In contrast, the increase of SA-β-gal-positive cells was canceled by knockdown of p53 (Figure 2F,G). Taken together, these results indicate that H3.3 depletion induces senescence-like phenotypes in a p53-dependent manner.

### 3.3. Identification of miR-22-3p as a Novel Regulator of H3.3

Next, we investigated the molecular mechanisms by which H3.3 was downregulated in senescence induction. Previously, we reported that miRNAs play important roles for senescence induction in human fibroblasts [25]. Therefore, we examined the possibility that the expression level of H3.3 is also regulated by miRNAs. To predict the candidate miRNAs that target H3.3 directly, we used two widely used algorithms, microRNA.org (accecced on 4 March 2015) and TargetScan [31,37,38]. These two algorithms revealed that miR-22-3p could directly bind *H3F3A* and *H3F3B* mRNA (Figure 3A).

To investigate the involvement of H3.3 and miR-22 during replicative senescence, we evaluated the expression of miR-22-3p in human fibroblasts. The expression levels of miR-22-3p in young, pre-senescent, and senescent TIG-3 cells were analyzed using the RT-qPCR method (Figure 3B). In agreement with our previous study [25], the expression level of miR-22-3p was upregulated during cellular senescence in TIG-3 cells. Since the expression levels of both *H3F3A* and *H3F3B* genes were significantly decreased during replicative senescence in human fibroblasts (Figure 1C), these results suggest that there is an inverse correlation between miR-22-3p and H3.3 expression.

To confirm that miR-22-3p inhibits H3.3 expression, we performed RT-qPCR and western blot analysis after the transfection of synthetic miR-22-3p mimic into TIG-3 cells. We found that miR-22-3p suppressed H3.3 at the protein level (Figure 3C). However, miR-22-3p suppressed only the expression of *H3F3B* genes at the mRNA level (Figure 3D). Contrary to the target prediction of microRNA.org and TargetScan, miR-22-3p induced the upregulation of *H3F3A.* Considering that miR-22-3p induced the downregulation of H3.3 at the protein level in TIG-3 cells, these data suggest that miR-22-3p directly targets *H3F3B* mRNA and inhibits the translation of H3.3 from *H3F3A mRNA*.

### 3.4. miR-22-3p Targets H3F3B Encoding Histone Variant H3.3

Finally, we examined whether H3F3B is directly suppressed by miR-22-3p using luciferase reporter construct containing the 3′-UTR of H3F3B mRNA (Figure 4A). The three putative binding sites (named regions 1, 2, and 3) for miR-22-3p on the 3′-UTR of H3F3B mRNA (Figure 4A) were predicted by miRNA.org in silico [33]. The full length of human H3F3B 3′-UTR, containing three putative binding sites, was amplified and cloned into pmirGLO vector. Additionally, a series of five mutant UTRs (Δsite regions 1 to 2, Δsite region 3, Δsite region 1, Δsite region 2, and Δsite regions 1 and 2) were generated, as shown in Figure 4B. Since the co-transfection efficacy of miRNA and its reporter construct in TIG-3 cells was quite low, we used 293T cells whose transfection efficacy is quite high for these experiments. In 293T cells, miR-22-3p significantly reduced the luciferase activities of the WT 3′-UTR reporter compared with the negative miR-control (Figure 4B). In contrast, neither Δsite regions 1 to 2 nor Δsites 1 and 2 were repressed by miR-22-3p, which indicates that miR-22-3p directly binds to these target sites (Figure 4B). Interestingly, single deletions of region 1 or region 2 were repressed by mir-22-3p, indicating a compensatory role of both target sites (Figure 4B).

## 4. Discussion

In the present study, we report for the first time that miR-22-3p can negatively regulate H3.3 abundance, which induces a p53-dependent senescence phenotype in fibroblasts, suggesting that H3.3 loss acts as a key molecular switch controlling cellular senescence. H3.3 is expressed throughout the cell cycle in various tissues and cell lineages, and is unevenly incorporated into chromosomes of cells [20]. Furthermore, H3.3 is enriched in highly transcribed genes, enhancers, promoters, telomeres and pericentric heterochromatins [18,19,20]. Recently, a study has reported that loss of H3.3 results in disruption of the normal structure and function of telomeres and centromeres, and results in anaphase bridges and lagging chromosomes, followed by inducing activation of the p53 pathway [39]. Abnormal mitosis results in G1 cell cycle arrest through p53-dependent activation of p21 in a DNA damage-dependent manner [40,41,42,43,44,45,46,47,48]. Thus, it appears that H3.3 function maintains chromosome integrity during the mitosis phase of the cell cycle. However, a recent study reported that chromosome mis-segregation in anaphase induces the phosphorylation of H3.3 at serine 31, and thereby activates p53 in a DNA damage-independent manner [49]. Hence, the relationship between H3.3 loss, H3.3 phosphorylation and p53 activation remains controversial.

On the other hand, H3.3 and the HIRA complex, a H3.3 histone chaperon, mediate the formation of SAHF in a replication-independent manner in replicative or Ras-induced senescent cells [17]. Some studies have reported that senescence-associated H3.3 and its tail cleavage product were increased during senescence, and deposited into the chromosome of senescent cells, and that ectopic H3.3 expression induces senescence [16,21]. However, in contrast to these studies, our study shows that the increased H3.3 or its cleavage product was not detectable by our immunoblotting method with the anti-H3.3 antibody used for the previous study [21] in replicative senescent cells, and that SAHF can be formed by H3.3 loss, suggesting that H3.3 is dispensable for SAHF formation. It remains unexplored whether the downregulation of H3.3 by miR-22-3p depends on cell type, cell-cycle, and various stressors involved in senescence.

Several studies also investigated the roles of histone proteins for senescence induction. In human fibroblasts, H2A.Z, which is a histone variant of H2A, is associated with p53/p21-dependent cellular senescence [50]. H2A.Z inhibits the p21 expression via regulating Tip60 histone acetyltransferase. Consistent with this report, we also observed downregulation of H2A.Z in cellular senescence. Other study reported that oncogene-induced senescence promotes the accumulation of H2A.J, which is a histone variant of H2A in human fibroblasts [51]. Since we observed the low expression of H2A.J both in young and senescent TIG-3 cells, the expression change of histone proteins might be dependent on the type of senescence. Considering our results and these reports, the expression change of histone proteins might be useful for the classification of cellular senescence based on the molecular subtypes.

Consistent with our previous findings [25], our study shows that miR-22-3p expression levels were elevated during replicative senescence. A study has reported that the expression of miR-22 is directly regulated by p53 [52]. In addition to miR-22-3p regulation by p53, the p53 pathway is also activated by H3.3 loss [39]. Thus, it is possible that the activation of p53 during the senescence program triggers H3.3 loss upon induction of senescence-associated miR-22-3p, and that senescence is induced and maintained by a p53-induced positive feedback loop between miR-22-3p induction and H3.3 loss. Most importantly, our study suggested that miR-22-3p induced and reinforced senescence by concomitantly regulating not only downstream of the senescence pathway but also epigenetic genes such as H3F3B, which are involved in the structure of chromatin in eukaryotic cells.

Recently, several studies have reported that H3.3 is overexpressed in human cancers [53] and that H3.3 mutations are identified at K27 and G34 of *H3F3A*-encoded H3.3 [54,55,56]. The expression of K27M-mutated H3.3 results in neoplastic transformation [57], suggesting that H3.3 mutations could play a role in cancer progression. However, in contrast, miR-22 expression is decreased in human cancer cells [25,52]. In addition, we previously demonstrated that synthetic miR-22 injection suppresses tumor growth and metastasis in a mouse model of breast cancer metastasis, suggesting a therapeutic use of miR-22 in metastasis [25]. Thus, it is possible that downregulation of H3.3 by miR-22-3p induces the senescence program that acts as a barrier to cancer initiation and progression. Therefore, selective inhibition of H3.3 might be a promising approach for cancer therapy.

For the development of miR-22-3p- or H3.3 siRNA-based therapeutics, it might be necessary to investigate whether miR-22-3p or H3.3 siRNA induces cellular senescence in epithelial cells, because, in most cases, tumor initiation occurs in epithelial cells [58]. In addition, it is also important to elucidate the mechanisms by which H3.3 downregulation induces p53/p21-dependent cellular senescence.

In conclusion, our study provides a novel insight into H3.3 function for cellular senescence which is regulated by miR-22-3p (Figure 5). Therefore, it might be important to further elucidate the histone functions to understand the molecular mechanisms underlying cellular senescence. Since senescence induction is a natural barrier to tumor onset and development, our results might contribute to the development of novel therapeutic approaches that induce the senescence-like phenotypes in transform and cancer cells.

## Figures and Tables

**Figure 1 genes-15-00543-f001:**
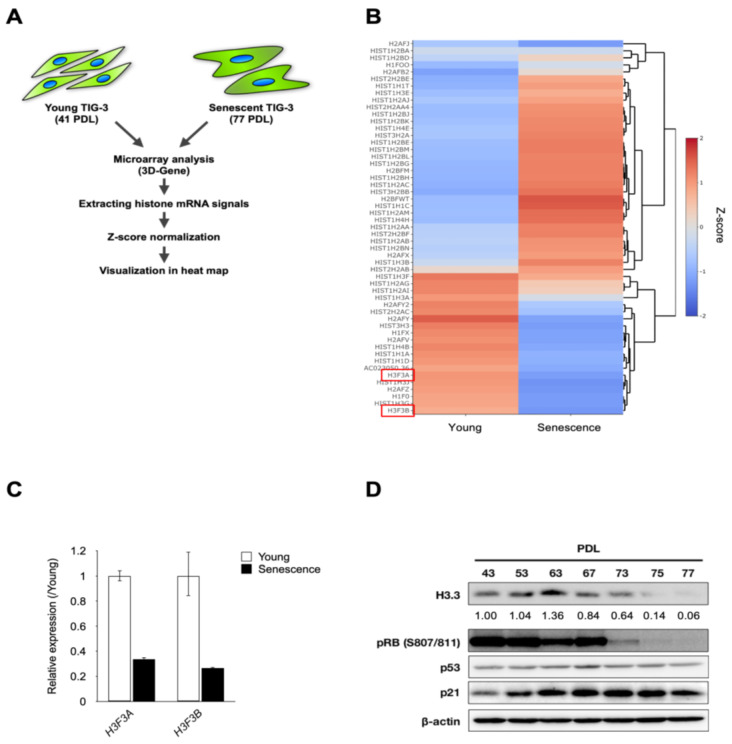
Histone variant H3.3 was decreased during replicative senescence in human diploid fibroblasts. (**A**) The scheme for microarray analysis using young and senescent TIG-3 cells. (**B**) Heatmap of histone mRNA expression between young and senescent TIG-3 cells. Each signal was retrieved from GSE162201. Color scale indicates the z-score. Red color indicates the gene has greater expression level than the mean. Blue color indicates the gene has lesser expression level that the mean. The red box indicates *H3F3A* and *H3F3B* gene (**C**) RT-qPCR analysis of *H3F3A* and *H3F3B* in young (42–44 PDL) and senescent (73–79 PDL) TIG-3 cells. (**D**) Western blotting analysis of H3.3, p53, p21, phosphorylation of RB (S807/811), and β-actin as internal control during replicative senescence. PDLs of TIG-3 cells are indicated above each line. Relative levels of H3.3 were normalized against the corresponding levels of β-Actin. Intensity was calculated using ImageJ.

**Figure 2 genes-15-00543-f002:**
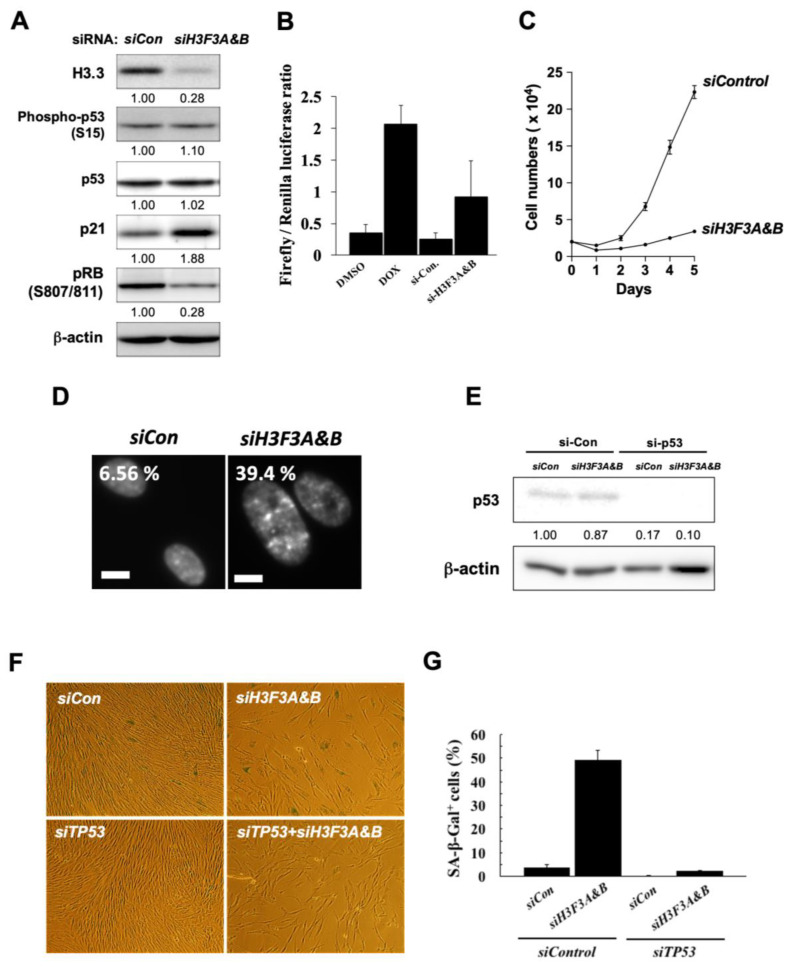
H3.3 depletion induced cellular senescence through p53/p21 activation. (**A**) H3.3 knocked down in TIG-3 cells transfected with siRNAs targeting *H3F3A* and *H3F3B.* On the third day after transfection with 10 nM of each siRNAs targeting *H3F3A* and *H3F3B*, the expression levels of H3.3, phosphorylation of p53, p53, p21, phosphorylation of RB (S807/811), and β-actin were analyzed by western blotting analysis. Relative levels of each protein were normalized against the corresponding levels of β-Actin. Intensity was calculated using ImageJ. (**B**) p21-promoter-driven luciferase reporter analysis. The firefly luciferase reporter vector containing p53-binding sites in p21 promoter region and renila luciferase reporter vector were co-transfected siRNAs targeting *H3F3A* and *H3F3B* or negative control siRNA into TIG-3 cells. The luciferase levels were normalized with relative light units (RLUs) of renilla luciferase activity in cells. As a positive control, doxorubicin (dox), a potent activator of p53, was treated at 5 μg/mL for 18 h with TIG-3 cells. (**C**) Cell growth of H3.3 knocked-down cells. Young TIG-3 cells were reverse-transfected with 10 nM of each siRNA targeting *H3F3A* or *H3F3B*, and the siRNA-transfected cells were counted at various times from 1 to 4 days after transfection. (**D**) SAHF formation in the H3.3-depleted cells. DAPI, 4′,6-diamidino-2-phenylindole, was used to visualize SAHF. Scale bar = 10 μm. (**E**) Validation of p53 knocked down in TIG-3 cells co-transfected with siRNAs targeting *H3F3A*, *H3F3B*, and *TP53*. The p53 and β-actin were analyzed by western blotting. Relative levels of p53 were normalized against the corresponding levels of β-Actin. Intensity was calculated using ImageJ. (**F**) SA-β-Gal activity assay in the p53 and/or H3.3 knockdown cells. SA-β-Gal-positive cells were stained on 7 days after transfection. (**G**) Quantification of SA-β-Gal-positive cells in (**F**).

**Figure 3 genes-15-00543-f003:**
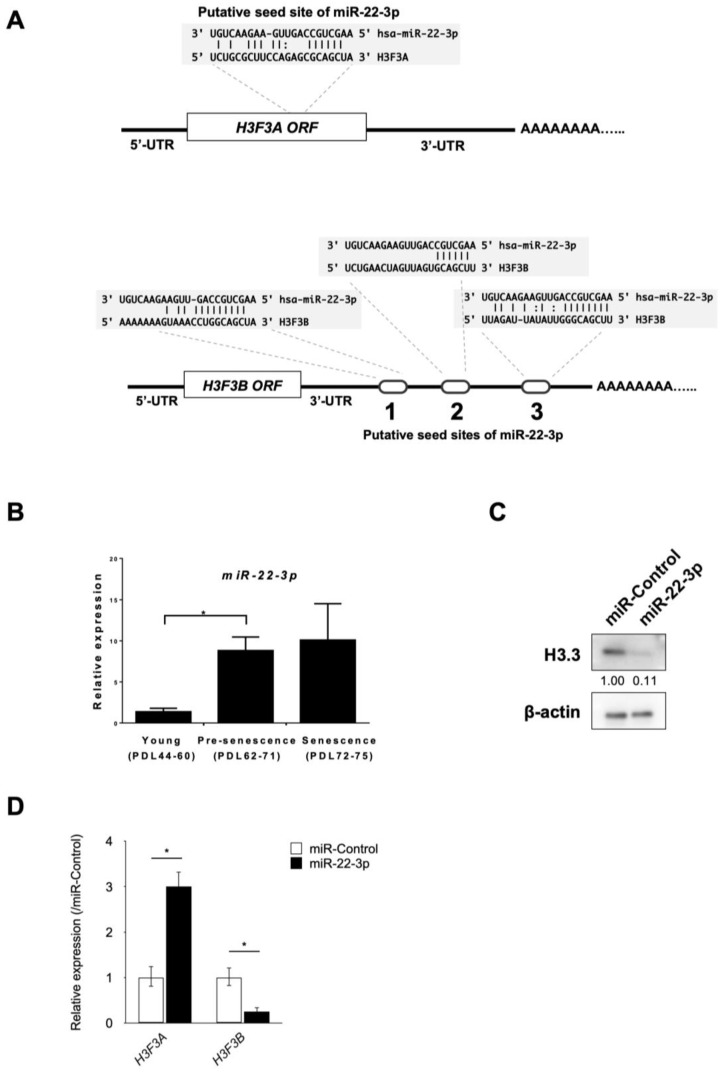
Identification of miR-22-3p as a novel regulator of H3.3. (**A**) Scheme of the putative miR-22-3p seed site in *H3F3A* and *H3F3B* genes. (**B**) RT-qPCR analysis of miR-22-3p in young, pre-senescence, and senescence stages of TIG-3 cells. Population doubling levels (PDLs) of TIG-3 cells are indicated in the graph. (**C**) Western blotting analysis of H3.3 and β-actin as internal control in young TIG-3 cells transfected with miR-22-3p mimic or control. The intensity of the band was normalized to β-actin and the values are shown below the panel as relative to miR-Control. (**D**) RT-qPCR analysis of *H3F3A* and *H3F3B* expression in young TIG-3 cells transfected with miR-22-3p mimic. * *p*-value < 0.05.

**Figure 4 genes-15-00543-f004:**
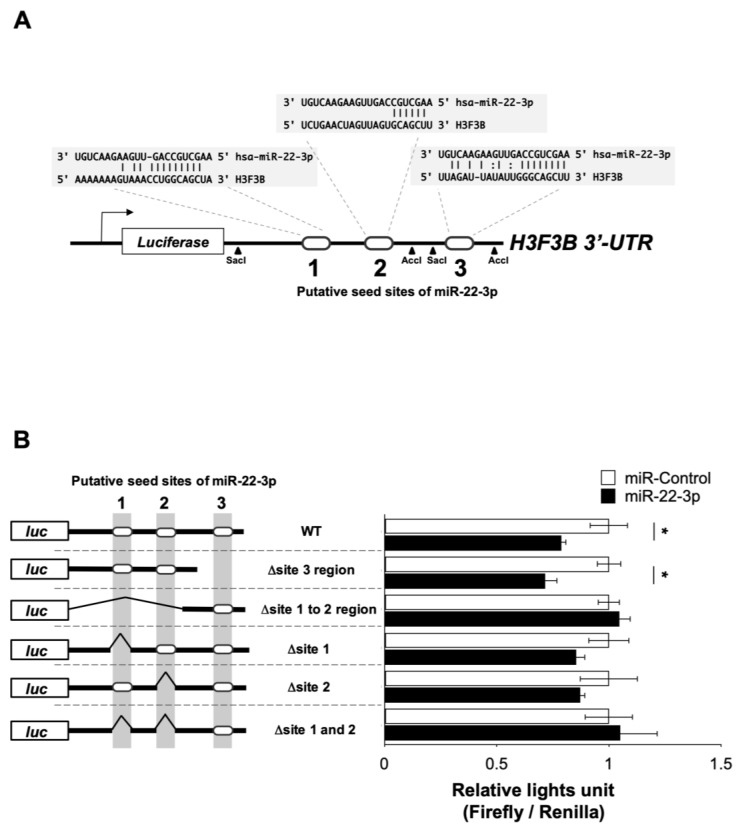
miR-22-3p targets H3F3B encoding histone variant H3.3. (**A**) Scheme of the luciferase reporter construct containing three putative miR-22-3p seed sites in 3′-UTR of *H3F3B* gene. The putative miR-22-3p seed sites are indicated as 1 to 3. To construct deletion mutants of luciferase vectors, cleavage sites of restriction enzymes, SacI and AccI, are indicated in the graph. (**B**) 3′-UTR luciferase reporter assay. The full-length of H3F3B 3′-UTR (WT) was constructed, and the deletion series, Δsite regions 1 to 2, Δsite region 3, Δsite 1, Δsite 2, and Δsite 1 and 2 were constructed. Each construct was co-transfected with miR-22-3p (black bar) or negative control miRNA (white bar) into 293T cells. The relative luciferase levels were normalized with RLUs of renilla luciferase activity in the cells transfected miR-22-3p or negative control miRNA and calculated from ratio of the normalized values in cells transfected with miR-22-3p and negative control miRNA.* *p*-value < 0.05.

**Figure 5 genes-15-00543-f005:**
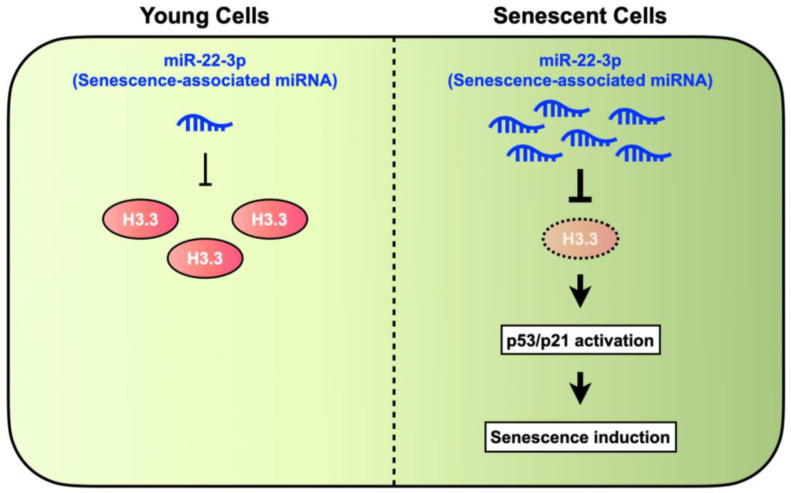
Summary of H3.3 roles in senescence induction. H3.3 was downregulated by miR-22-3p, which leads to p53/p21 pathway-dependent cellular senescence.

## Data Availability

Data are contained within the article.

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
