# Peer review of "Downregulation of Histone H3.3 Induces p53-Dependent Cellular Senescence in Human Diploid Fibroblasts"

_genes, 2024, doi:10.3390/genes15050543_

Round 1
Reviewer 1 Report
Comments and Suggestions for Authors
This paper reports a study looking at molecular consequences of H3.3 depletion in senescent fibroblast cells. Although the paper is well written and of high interest with an appropriate methodology, there are some major concerns with this submission:
1) The figures were not included in the result section (or anywhere else in the manuscript). Hard to visualise what the researchers have done without the visualisation of the results.
2) The discussion is quite limited (and mainly focused on what the authors have reported before) and should be significantly extended including other genes regulated by mi-22-3p (for example eukaryotic translation initiation factor 4G (eIF4G) for example) and /or histone protein pathways involved in the context of ageing.
The authors should further consul the guidelines for authors regarding the format of the manuscript including Figures through the adequate sections (and place the legend with the related figures, not at the end of the manuscript)
Author Response
Reviewer Responses
Reviewer#1
This paper reports a study looking at molecular consequences of H3.3 depletion in senescent fibroblast cells. Although the paper is well written and of high interest with an appropriate methodology, there are some major concerns with this submission:
- The figures were not included in the result section (or anywhere else in the manuscript). Hard to visualise what the researchers have done without the visualisation of the results.
Response to the comment#1
We sincerely apologize for not attaching any results to the manuscript. In the revised manuscript, we inserted our results into the Result section as Figure 1-5.
- The discussion is quite limited (and mainly focused on what the authors have reported before) and should be significantly extended including other genes regulated by mi-22-3p (for example eukaryotic translation initiation factor 4G (eIF4G) for example) and /or histone protein pathways involved in the context of ageing.
Response to the comment#2
We appreciate the insightful comments and discussed histone expression involved in senescence at the Discussion section (page 11, line 337 to 347)
Page. 11, line 337 to 347:
Several studies also investigated the roles of histone proteins for senescence induction. In huma fibroblasts, H2A.Z which is a histone variant of H2A is associated with p53/p21-dependent cellular senescence [50]. H2A.Z inhibits the p21 expression via regulating Tip60 histone acetyltransferase. Consistent with this report, we also observed down-regulation of H2A.Z in cellular senescence. Other study reported that onco-gene-induced senescence promotes the accumulation of H2A.J which is a histone variant of H2A in human fibroblasts [51]. Since we observed the low expression of H2A.J both in young and senescent TIG-3 cells, the expression change of histone proteins might be dependent on the type of senescence. Considering our results and these reports, the expression change of histone proteins might be useful for the classification of cellular senescence based on the molecular subtypes.
- The authors should further consul the guidelines for authors regarding the format of the manuscript including Figures through the adequate sections (and place the legend with the related figures, not at the end of the manuscript)
Response to the comment#3
According to the sincere consideration of reviewer#1, we carefully checked our revised manuscript with the author instruction.

Reviewer 2 Report
Comments and Suggestions for Authors
Comments to the authors:
The paper titled "Down-regulation of histone H3.3 induces p53-dependent cellular senescence in human diploid fibroblasts" by Yamamoto et al. presents novel insights into the roles of histone variant H3.3 and microRNA miR-22-3p in the induction of cellular senescence, a mechanism that acts as a barrier to cancer initiation and progression. Overall, this paper is exciting and I have some questions that need to be addressed by the authors.
1: The introduction provides a thorough background on cellular senescence and the role of histone alteration. However, it could benefit from a more detailed discussion of the specific functions and previous findings related to H3.3 in cellular senescence, to better contextualize the novelty and significance of the current findings.
2: The methods used for cell culture, RT-qPCR, DNA microarray, Western blotting, luciferase reporter assay, and other experimental techniques are described adequately, but the paper could provide more details about the controls used in each experiment, especially in the luciferase reporter assays and siRNA knockdown studies, to ensure readers understand the specificity and efficacy of these approaches.
3: The results section is comprehensive, presenting data on H3.3 downregulation, the role of miR-22-3p, and the involvement of the p53 pathway in cellular senescence. However, the paper would benefit from additional quantitative data analysis and statistical validation to strengthen the conclusions drawn. For example, quantifying Western blot results and a more detailed analysis of the luciferase reporter assay results would add robustness to the findings.
4: The discussion thoughtfully interprets the findings within the broader context of histone biology, cellular senescence, and cancer biology. Nevertheless, it could be enhanced by a deeper exploration of the potential implications of H3.3 downregulation and miR-22-3p regulation in cancer therapy and aging. Additionally, discussing potential limitations of the study and areas for future research would provide a more balanced view.
5: The paper includes a comprehensive list of references, supporting its claims and providing a solid background for readers unfamiliar with the topic. It would be helpful to ensure that all recent and relevant studies, mainly those published right before this study, are included to reflect the current state of knowledge in the field.
6: The figures provided are informative and contribute to understanding the experimental findings. However, improving the resolution and clarity of some figures and possibly including supplementary figures or tables that detail the experimental conditions could enhance the reader's ability to interpret the data.
7: The paper makes a significant contribution to the field by elucidating a novel regulatory mechanism of cellular senescence involving H3.3 and miR-22-3p. Emphasizing the novelty and potential applications of these findings in the abstract and conclusion sections would help highlight the importance of the work.
In summary, while the paper presents significant findings that advance our understanding of the molecular mechanisms underlying cellular senescence, enhancements in the areas mentioned could improve its clarity, impact, and usefulness to the field.
Comments on the Quality of English Language
Minor editing of the English language required
Author Response
Reviewer Responses
Reviewer#2
The paper titled "Down-regulation of histone H3.3 induces p53-dependent cellular senescence in human diploid fibroblasts" by Yamamoto et al. presents novel insights into the roles of histone variant H3.3 and microRNA miR-22-3p in the induction of cellular senescence, a mechanism that acts as a barrier to cancer initiation and progression. Overall, this paper is exciting and I have some questions that need to be addressed by the authors.
- The introduction provides a thorough background on cellular senescence and the role of histone alteration. However, it could benefit from a more detailed discussion of the specific functions and previous findings related to H3.3 in cellular senescence, to better contextualize the novelty and significance of the current findings.
Response to the comment#1
According to the insightful comments, we discussed the roles of H3.3 for cellular senescence in the Introduction (page. 2, line 49 to 53) and Discussion section (page. 11, line 337 to 347) as described below.
Page 2, line 49 to 53:
In addition, Pak2 induced HIRA mediated nucleosome assembly of H3.3 in human fibroblast during oncogenic induced senescence [22]. On the other hands, loss of CDH1 reduced H3.3 levels in brain chromatin, which leads to lifespan shortening [23]. While a number of studies report the association of H3.3 expression with cellular senescence and aging, it still remains unclear the roles of H3.3 in cellular senescence.
page. 11, line 337 to 347:
Several studies also investigated the roles of histone proteins for senescence induction. In huma fibroblasts, H2A.Z which is a histone variant of H2A is associated with p53/p21-dependent cellular senescence [50]. H2A.Z inhibits the p21 expression via regulating Tip60 histone acetyltransferase. Consistent with this report, we also observed down-regulation of H2A.Z in cellular senescence. Other study reported that oncogene-induced senescence promotes the accumulation of H2A.J which is a histone variant of H2A in human fibroblasts [51]. Since we observed the low expression of H2A.J both in young and senescent TIG-3 cells, the expression change of histone proteins might be dependent on the type of senescence. Considering our results and these reports, the expression change of histone proteins might be useful for the classification of cellular senescence based on the molecular subtypes.
- The methods used for cell culture, RT-qPCR, DNA microarray, Western blotting, luciferase reporter assay, and other experimental techniques are described adequately, but the paper could provide more details about the controls used in each experiment, especially in the luciferase reporter assays and siRNA knockdown studies, to ensure readers understand the specificity and efficacy of these approaches.
Response to the comment#2
We apologize the inadequate information about the materials and methods. We added the additional information about the luciferase reporter assay and siRNA knockdown at the Materials and Methods section (page.3, line 125 to 127 and 137 to 139) as below.
Page 3, line 125 to 127:
pmirGLO vector has one insertion site of miRNA target sequence at the 3’ UTR of the firefly luciferase gene. Renilla luciferase gene is also coded into pmirGLO vector for normalization.
Page 3, line 137 to 139:
For 3’-UTR luciferase reporter assays, each construct (100 ng) was co-transfected with miR-22-3p mimic or miR-control mimic (mirVana miRNA, Thermo Fisher Scientific) into a 96-well plate with DharmFECT Duo transfection reagent (Thermo Fisher Scientific).
- The results section is comprehensive, presenting data on H3.3 downregulation, the role of miR-22-3p, and the involvement of the p53 pathway in cellular senescence. However, the paper would benefit from additional quantitative data analysis and statistical validation to strengthen the conclusions drawn. For example, quantifying Western blot results and a more detailed analysis of the luciferase reporter assay results would add robustness to the findings.
Response to the comment#3
According to the insightful comments, we calculated the results of immunoblot analysis using ImageJ software (Fig. 1D, 2A, 2E, 3C). In addition, we analyzed the results of reporter assay in Fig. 2B and confirmed the significant difference between control siRNA and siRNA against H3F3A&B (Page 5, line 209 to 210).
- The discussion thoughtfully interprets the findings within the broader context of histone biology, cellular senescence, and cancer biology. Nevertheless, it could be enhanced by a deeper exploration of the potential implications of H3.3 downregulation and miR-22-3p regulation in cancer therapy and aging. Additionally, discussing potential limitations of the study and areas for future research would provide a more balanced view.
Response to the comment#4
We agree with the reviewer’s comments, we additionally discussed the potential implications of H3.3 downregulation and miR-22-3p regulation in cancer therapy and aging as below (page 11, line 365 to 367). We also explained the potential limitations and perspectives of our study (page 12, line 368 to 372).
Page 11, line 365 to 367:
Thus, it is possible that down-regulation of H3.3 by miR-22-3p induces senescence program that acts as a barrier to cancer initiation and progression. Therefore, selective inhibition of H3.3 might be a promising approach for cancer therapy.
Page 12, line 368 to 372:
For the development of miR-22-3p or H3.3 siRNA based therapeutics, it might be necessary to investigate whether miR-22-3p or H3.3 siRNA induces cellular senescence in epithelial cells. Because, in most cases, tumor initiation is occurred in epithelial cells [58]. In addition, it is also important to elucidate the mechanisms by which H3.3 downregulation induces p53/p21 dependent cellular senescence.
- The paper includes a comprehensive list of references, supporting its claims and providing a solid background for readers unfamiliar with the topic. It would be helpful to ensure that all recent and relevant studies, mainly those published right before this study, are included to reflect the current state of knowledge in the field.
Response to the comment#5
According to the insightful comments, we added key references regarding cellular senescence in the Introduction section (Page 1, line 27 to 33) as described below.
Page 1, line 27 to 33:
Cellular senescence is a physiological and homeostatic processes with permanent cell growth arrest and induced by a wide range of cellular stress such as DNA damage, oxidative stress, and oncogene activation [1-3]. Senescent cells are characterized with flattened morphology, heterochromatic changes, loss of Lamin-B1 and, expression of senescence-associated ß-galactosidase (SA-βGal). In most cases, tumor suppressor genes such as p53/p21 and p16 are activated in senescent cells. Therefore, Cellular senescence is considered to act as a natural barrier to tumorigenesis [1-6].
- The figures provided are informative and contribute to understanding the experimental findings. However, improving the resolution and clarity of some figures and possibly including supplementary figures or tables that detail the experimental conditions could enhance the reader's ability to interpret the data.
Response to the comment#6
We apologize for the low resolution of the figures. In the revised manuscript, we improved the resolution and clarity of all the figures.
- The paper makes a significant contribution to the field by elucidating a novel regulatory mechanism of cellular senescence involving H3.3 and miR-22-3p. Emphasizing the novelty and potential applications of these findings in the abstract and conclusion sections would help highlight the importance of the work.
Response to the comment#7
According to the insightful comments, we emphasized the novelty and potential applications of our findings in the Abstract and Conclusion section described below.
Page 1, line 20 to 23:
Therefore, our results reveal for the first time the molecular mechanisms for cellular senescence which are regulated by H3.3 abundance. Taken together, our studies suggest that H3.3 exerts functional roles in regulating cellular senescence and is a promising target for cancer therapy.
Page 12, line 373 to 378:
In conclusion, our study provides the novel insight into H3.3 function for cellular senescence which is regulated by miR-22-3p. Therefore, it might be important to further elucidate the histone functions for understanding the molecular mechanisms underlying cellular senescence. Since senescence induction is a natural barrier to tumor onset and development, our results might contribute to the development of novel therapeutic approaches that induce the senescence-like phenotypes in transform and cancer cells.

Round 2
Reviewer 1 Report
Comments and Suggestions for Authors
Authors have addressed my concerns thanks
Reviewer 2 Report
Comments and Suggestions for Authors
The authors have addressed all the concerns